# Understanding Dielectrics: Impact of External Salt Water Bath

**DOI:** 10.3390/ma12122033

**Published:** 2019-06-25

**Authors:** Jonathan Phillips, Alexander Roman

**Affiliations:** Mechanical and Aerospace Engineering, Naval Postgraduate School, Monterey, CA 93950, USA; ajroman@nps.edu

**Keywords:** dielectrics, capacitors, capacitor theory

## Abstract

As predicted by the theory of super dielectric materials, simple tests demonstrate that dielectric material on the outside of a parallel plate capacitor dramatically increases capacitance, energy density, and power density. Simple parallel plate capacitors with only ambient air between the plates behaved as per standard theory. Once the same capacitor was partially submerged in deionized water (DI), or DI with low dissolved NaCl concentrations, still with only ambient air between the electrodes, the capacitance, energy density, and power density, at low frequency, increased by more than seven orders of magnitude. Notably, conventional theory precludes the possibility that material outside the volume between the plates will in any fashion impact capacitive behavior.

## 1. Introduction

In this paper a novel experiment was conducted to test further a new theory of dielectrics, the so-called super dielectric material (SDM) theory. The experimental design of this work was intended to provide a contrast between conventional dielectric theory, as presented in physics texts, and SDM theory. That is, the experiments were designed such that the outcome could only be consistent with one of these theories.

The basic arguments of the SDM theory are not widely disseminated, hence there is value in a brief review. To wit: The central hypothesis of the SDM theory is that dielectrics increase capacitance by polarizing opposite to the polarity of charges on the electrodes. This can be understood from a five-part argument [1,2]. (1) Dielectric material polarizes in the opposite direction to any field applied to it. This occurs because the positive charge in a dielectric moves toward the negative electrode and negative charge moves toward the positive electrode. (2) Placed between the electrodes of a standard parallel plate capacitor, the dielectric material creates a field opposite in direction to the electric field created by charges on the electrodes, in all space, not just the region between the plates. (3) As the field at any point in space is the vector sum of the fields of all individual charges, the dielectric in a parallel plate capacitor reduces the field, at all points, created by charges on the electrodes. (4) As “voltage”, a state property, is the scaler line integration of the electric field, and the dielectric reduces the field at all points, the dielectric necessarily reduces the “voltage” between any two points, including any path from infinity to an electrode. (5) It follows that as in the presence of a dielectric it takes more charge on the electrodes to reach a given capacitor voltage, dielectrics increase the electrode charge/voltage ratio. Thus, by definition, dielectrics increase capacitance.

There are some inherent predictions of the SDM model. One example is the prediction that the effectiveness of a dielectric is the product of the length of charge separation within it (dipole length), and the density of charges (dipole density). This was tested and found accurate in earlier work [3,4]. Other work showed that, as predicted by the model, high dielectric constants would be found for salt water saturated fabric [5], for salt water saturated nano-tubes on the surface of anodized titania [6], for porous solids saturated with various salts [7,8], for fumed silica gels containing salt water [9], for water saturated with salts other than NaCl such as KOH and NH_3_Cl [10]. Other studies show that both metal and carbon can be used as the electrode material [2], and for non-aqueous polar fluids such as DMSO containing dissolved salts, etc. Each of the existing SDM studies [1,2,3,4,5,6,7,8,9,10,11,12,13] adds to the corpus of data supporting the SDM hypothesis.

Another prediction of the model is that any mechanism that reduces the field at all points in space around a capacitor will increase the capacitance. This implies that in a standard parallel plate capacitor dielectric material need not be between the plates in order to impact capacitance. Indeed, dielectric material outside the volume between the electrodes should, under correct circumstances, increase capacitance. Consistent with this prediction of SDM theory, our team recently demonstrated that a parallel plate capacitor with high dielectric material only outside the volume between the plates acts “as if” there is a high dielectric constant material between the plates [2]. As shown in that study, a simple capacitor composed of titanium electrodes and a thin plastic dielectric had, as anticipated, a very low capacitance. Second, the control capacitor, still the same plastic dielectric, was modified on its outside only. Specifically, it was covered on the outside in a continuous thin layer (<1 mm thick) of a particular gel type super dielectric material. This increased measured capacitance by as much as seven orders of magnitude higher than the control below ~1 V. This finding is consistent with SDM theory, and completely contrary to standard theory.

Further experiment is needed to demonstrate the generality of the SDM hypothesis, particularly as it applies to the “dielectric on the outside” prediction. In the present study simple parallel plate capacitors containing only air in the volume between the titanium foil plates were (1) immersed in air, (2) immersed in distilled deionized water (DI), (3) immersed in deionized water containing 0.5% NaCl (8.5 × 10^−2^ mol/L). The results confirm SDM predictions regarding the efficacy of “dielectric on the outside”. That is, contrary to standard theory that immersion in any material cannot impact measured capacitance, it was found, in agreement with the SDM model, that immersion in salt water increased capacitance by more than seven orders of magnitude.

In addition to these experiments, studies of the behavior of (1) DI, and (2) DI containing ~5.0 wt % NaCl the dielectric between the electrodes is presented. The finding that the dielectric constant of DI water is remarkably high, in fact >10^9^ in particular circumstances, confirms earlier studies showing pure water, at low frequency, has a remarkably high dielectric value [14,15]. These results suggest, according to SDM theory, that well organized dipole formation must occur in water exposed to electric fields, and suggest there is value in continuing research on the dielectric behavior of water.

## 2. Experimental

Two different parallel plate capacitors, with electrodes made of Ti sheets (0.1 mm thick) 3 cm × 3 cm, covering an air gap of dimension 2.5 cm × 2.5 cm were employed. The only difference between the two was the size of the gap between the electrodes: 6 mm in one case and 20 mm in the latter. As shown in Figure 1, the Ti sheets were held between materials known to have low (<100) dielectric constants; rubber layer to grip the titanium sheets, and the gap created by layers of clear Acrylic sheet.

Several different capacitor configurations were studied. In all cases two different dielectric must be specified; an inner dielectric, that is the dielectric material between the electrodes and an outer dielectric, that is the dielectric material surrounding/outside the volume between the electrodes. The distance between electrodes was also specified below, because capacitors were virtually identical but for the distance between electrodes were studied. Specifically, the behavior observed for a capacitor in which the electrode distance was 20 mm (20 mm capacitor) was contrasted with one for which the electrode separation was 6 mm (6 mm capacitor).

### 2.1. Control

In the control cases the capacitor was simply placed on the lab bench in the ambient air (AIR). Both the inner and the outer dielectric were ambient air. There were two controls: One in which the titanium sheet electrode separation was 20 mm and one in which it was 6 mm.

### 2.2. Dielectric on Outside

In the “dielectric on the outside” configuration (DOC), the inner dielectric was the same as in the control case; ambient air. The outer dielectric was a super dielectric material, either DI or DI with dissolved NaCl, generally 0.5% by weight. The bath surrounding the capacitor in all cases was about 500 cm^3^. In the DOC configurations ~95% of the electrode surface was covered in liquid. The remainder was in the ambient environment. Two cases were studied: (1) The capacitor was partially submerged in DI water (DI-DOC), or (2) The capacitor was partially submerged in DI water containing dissolved NaCl (S-DOC), that is salt water.

### 2.3. Parameter Computation

The fact that the dielectric is on the “outside” leads to a conundrum in terms of computing and labeling parameters. That is, the standard nomenclature requires a volume, and that volume is always assumed to be that of the dielectric “inside” the electrodes. To address this conundrum the computations were conducted “as if” only the volume between the electrodes is contributing, and the resulting values are called “effective dielectric constant”, and “effective energy density”.

### 2.4. Dielectric on the Inside

In the distilled water-dielectric on the inside configuration (DI-DIC) distilled water was used to fill the space between the electrodes, which is the inner dielectric. The capacitors were placed on the lab bench, hence the outer dielectric was simply ambient air. In essence this is the standard geometry for testing the dielectric properties of a material. In the salt water-dielectric on the inside configuration (S-DIC) salt water, generally DI water containing 0.5 wt % dissolved NaCl, was used to fill the space between the electrodes, hence salt water is the inner dielectric. The capacitors were placed on the lab bench, hence, again, the outer dielectric was simply ambient air.

### 2.5. Testing Protocol

All data, dielectric constant, energy, and power density, were computed from the constant current discharge leg of charge/discharge cycles collected using a programmable galvanostat (BioLogic Model SP 300 Galvanostat, Bio-Logic Science Instruments SAS, Claix, France). Notably, the device is regularly tested by using it to measure the marked capacitance of both commercial supercapacitors and electrostatic capacitors. The agreement with nominal capacitance is always excellent. The device, in constant current discharge mode, was operated over the voltage range, 0 to 10 V. The rate of electrolysis of water was minimal at these voltages, insignificant bubble formation even after twelve hours of continuous running. Capacitance is defined in constant current to be:(1)$$C=\frac{I}{\frac{dV}{dt}},$$
where *C* is capacitance, *I* is current, *V* is voltage, and *t* is time. Clearly, if capacitance is not a function of voltage, voltage should decline linearly with time. As noted below and elsewhere [1], this is not always the case, particularly at “higher” frequencies.

The constant current method has advantages relative to more commonly employed methods for measuring capacitance. Constant current data is far easier to deconvolute than that obtained with cyclic voltammetry [16,17]. The constant current method also provides direct measures of energy and power density. In contrast, impedance spectroscopy [1,18,19,20] is limited to providing values based on measurements conducted over a very small voltage range, ±15 mV, thus is clearly not able to directly measure energy or power. In impedance spectroscopy a voltage independent capacitance (ideal) also is assumed; although, it is clear from a review of the literature that this is generally only true at a very low frequency [1]. For the capacitors studied in this work, as with most capacitors, the “ideal” behavior was not observed.

Capacitance is generally used to compute dielectric constant (*ε*) by Equation (2) for a parallel plate capacitor. This is the mathematical expression of the standard theory of dielectrics applied to parallel plate capacitors:(2)$$\varepsilon =\frac{C\;\times \;t}{A\;\times \;\varepsilon_{0}},$$
where *t* is the thickness of the dielectric layer, *A* is the area of the electrode, and *ε*_0_ is the permittivity of free space [21,22,23]. Equation (2), that is the standard theory of dielectrics applied to parallel plate capacitors, is based on the assumption that only the dielectric material between the electrodes contributes to the capacitance. This was clearly demonstrated to be an incorrect assumption in the present study, and an earlier study by our team [2]. Thus, following the precedent set in earlier work, dielectric constant, energy density, and power density were computed/reported below “as if” the only volume of significance is that between the plates. Energy was computed as the integral of the area under the voltage time data (V × s) multiplied by current (amps), and power was computed as the total energy of the discharge divided by the total discharge time.

On the discharge leg, two distinguishable ranges of capacitance as a function of voltage were found. In the first range from 10 V to ~1.2 V the capacitance was relatively low and not a subject of significant inquiry in this study. The capacitance and dielectric values reported were only reported based on data for the discharge between ~1.0 and 0 V. Over this range the voltage vs. time relationship was nearly linear in all cases for discharge times greater than ~1 s, indicating constant capacitance over this voltage region.

The standard protocol for testing involved three steps. The first step was charging to 10 V, generally at 1.5 mA. The second step was to hold the voltage for a period of time, for example 200 s. All parameters were derived from the third step, discharge of the capacitor from 10 V to 0 V at a constant current. Next, the polarity was reversed in all cases, and a mirror “negative” voltage studied. Thus, the capacitor was charged quickly to −10 V, held at that voltage for the same time as during the positive voltage sequence (e.g., 200 s) and then discharged to zero volts at the same current as the positive voltage discharge step. Generally, the reported values of parameters are the average of four cycles (circa Figure 2). In many cases, after four cycles, the value of the discharge current was changed, and the process repeated with the charge step, voltage, and voltage hold times unchanged. Changing the discharge voltage is the only means to change the discharge period/”frequency”. This permits an approximate analysis of the impact of frequency. Note: This three-step protocol is very similar to that employed to characterize the capacitance of commercial supercapacitors [24,25].

## 3. Results

The experiments were designed to collect capacitance, and “effective” values of dielectric constant, energy, and power density. This data was then employed for several purposes: (1) To validate the SDM hypothesis. Specifically, dielectric material outside the volume between the electrodes significantly impacts all capacitor performance parameters. (2) To provide a check of earlier studies indicating that distilled water has a remarkably high dielectric value at low frequency (ca. 1 Hz). (3) To determine if these parameters impact capacitor behavior: Maximum charging voltage, hold time, discharge current, salt concentration, and electrode separation distance.

### 3.1. Control

The discharge time, given the smallest allowed discharge current, for the galvanostat connectors simply placed just above the bench in ambient conditions and that obtained when the electrodes are connected to the capacitor in the AIR configuration are the same. The charging current shows the same pattern as well. This indicates that the galvanostat is not able to measure discharges that occur more rapidly than 5 × 10^−4^ V/s as this is the current an instrument leakage minimum. Thus, the measurements made for this study confirm that the capacitance was extremely low for the AIR configuration, but the measurement method employed was not sufficient to determine the actual capacitance. Assuming the standard dielectric constant for “air”, approximately 1, yielded a capacitance of 2 × 10^−13^ Farads (F) for the 20 mm separation capacitor, and 1.9 × 10^−12^ F for the 6 mm separation capacitor. In contrast, the capacitance measured below 1 V for the S-DOC 20 mm capacitor was ~4.5 × 10^−3^ F (discharge current 0.02 mA) and 9 × 10^−3^ F for the S-DOC 6 mm capacitor, or more than eight orders of magnitude higher than the AIR configuration in both cases.

### 3.2. Raw Data Outside Configuration

In Figure 2 the results for the DI-DOC of the 20 mm capacitor is illustrated with the raw data. The discharge time, on the order of three seconds from 1 V to 0 V, was many orders of magnitude higher than that observed in the control studies (<0.5 ms) of the same capacitor sitting in ambient air.

One key result was that the hold time had almost no impact on the discharge time, a result dramatically different from that observed for salt water. That is, the discharge time for a ten second, a two hundred second, and a six hundred second discharge were not distinguishable.

The behavior pattern of the S-DOC shared some aspects with the DI-DOC configuration, but also showed fundamental differences. An example of the former was the discharge shape (Figure 3). Discharge to about 2 V in all cases took place in less than five seconds, and then in some cases (e.g., long hold times) slowed dramatically. An example of the latter was the impact of hold time. In the case of DI-DOC the hold time at 1 s and 600 s was nearly equal, whereas for the S-DOC hold time had a considerable impact. As shown in Figure 3, the S-DOC the discharge time for a hold time of 600 s was 35× longer than for a hold time of 1 s.

### 3.3. Dielectric Values

In Figure 4 the effective dielectric constant below one V (20 mm electrode separation) for three different salt concentration (DI-DOC and S-DOC) of the outer dielectric, with ambient air, all cases, as the inner dielectric. Clearly the S-DOC configurations had higher effective dielectric values than the DI-DOC configuration, but it was also clear that the DI-DOC was displaying effective dielectric values at least five orders of magnitude higher than the classically reported dielectric value for water, ~80 [26]. These high values of the dielectric constant for DI at a low frequency/long discharge period were similar to those reported elsewhere [14,15] for distilled water.

Figure 4 also indicates that the effective dielectric constant for S-DOC was a function of the dissolved salt concentration. For example, the effective dielectric constant for a 250 s discharge of the 5 wt % NaCl solution was about 7× larger than for the 0.5 wt % NaCl solution.

Finally, Figure 4 indicates that the dielectric constant for discharge times greater than ~0.5 s were relatively constant, given all other protocol parameters constant. This suggests an effective “saturation” limit, where saturation in this study meant that the number of charges released through the circuit, that is the capacitance, was not impacted by current levels/discharge time. The finding that dielectric values were relatively flat as a function of discharge current, was not consistent with previous studies of SDM [1,3,4,5,6,7,8,9,10,11,12,13] on the “inside”. The physical basis for saturation of a dielectric was postulated to relate to full alignment of the dipoles in the dielectric. That is, at a particular voltage all the dipoles in the material were fully aligned, hence further increasing the voltage on the electrodes had no impact on the field generated by the dielectric [1,27], hence increasing voltage above the saturation voltage did not increase the amount of charge on the electrodes.

It was also clear that not all data was reasonably fit with a power law curve. The data for the 0.5 wt % NaCl case was nearly flat above a discharge time of 2.5 s, and clearly fell sharply for faster discharges. This was a “trend”, albeit very non-linear. In general, the reader should note that the power law curves fitted the data imperfectly, thus extrapolation of the fit curves did not provide quantitative prediction. Still, the finding of complex “trends” in a few cases did not detract from the primary message of the paper: Immersing a parallel plate capacitor in DI or low salt solution dramatically increased capacitance.

The value of the dielectric constant, remarkably high in all cases, was found to be a function of the electrode separation. Specifically, it was found that the dielectric constant was consistently higher for an electrode separation for 6 mm than it was for a separation of 20 mm (Figure 5). It was also found that the dielectric constant for salt water in the S-DOC was consistently higher than for the S-DIC configuration both for the 6 mm capacitor (shown) and the 20 mm capacitor.

### 3.4. Energy Density

In Figure 6, quantitative plots of energy density for dielectric “outside” configurations of the 20 mm capacitor at different salt levels are shown. Note that all data was in terms of “effective” values. That is, only the volume between the plates was employed as the volume in computations, yet it was clear that dielectric outside this volume was dramatically impacting the results. Although it was clear that the energy density of the S-DOC were higher than those of the DI-DOC, the trends suggest that for very slow discharges the energy densities for all salt levels might converge.

Similar broad trends in energy density were found for both the 6 mm and the 20 mm capacitors (Figure 7). Indeed, for the 6 mm capacitor energy density was highest for salt water (0.5 wt % all cases) on the outside (triangles), and in all equivalent cases, only salt concentration modified, the energy density was higher for salt water than for DI. The 6 mm capacitor consistently had higher energy density than the 20 mm capacitor in all equivalent configurations. This result was anticipated as in both the SDM and standard model of parallel plate capacitors energy density was inversely proportional to the electrode distance squared. In this study, the effective dielectric constant for salt water on the outside also increased as the electrode distance was reduced. This is another reason the increase in energy density with a decrease in electrode separation, was anticipated. It was also clear that the S-DOC pattern (not a clear trend for either data of these data sets) in energy density for longer discharge times (>10 s) was remarkably similar for the 20 mm and 6 mm capacitors.

As noted for other parameters, given the poor fit of some of the power law curves, quantitative extrapolation was not valid. Note: For the two “DOC” configurations shown the energy density was the effective energy density.

### 3.5. Power Density

In contrast to energy density, for all reported SDM based capacitors [1,2,3,4,5,6,7,8,9,10,11,12,13], power increases as the discharge time decreases. This indicates that for SDM based capacitors energy released during discharge is decreasing less quickly than the discharge time. This was also found true in the present study of SDM on the outside (Figure 8). As anticipated, with all other parameters constant, salt significantly also increased the power density; the power produced by S-DOC was at least an order of magnitude higher than equivalent DI-DOC at all discharge rates. Yet, it was also clear that DI-DOC performed extremely well.

## 4. Discussion

The mathematics employed in standard dielectric theory indicates an implicit assumption: The nature of the material on the ‘outside’ of a capacitor is irrelevant. A good example is the mathematics of the most ubiquitous capacitor, a parallel plate capacitor. To determine the dielectric constant of a material that fills the space between the electrodes of a parallel plate capacitor three values are required: Measured capacitance, the area of the electrodes, and the distance between them (Equation (2)). There is no mathematical provision made to account for the properties of material not between the electrodes. True also: In standard narrative descriptions of the impact of dielectrics on capacitance there is never consideration given to properties of material outside the volume enclosed by the electrodes. In contrast, in SDM theory the properties of all dielectric materials, both between the plates and outside the plates, must be considered. One notable shortcoming of the SDM theory is that there is no simple equation linking geometric and materials properties equivalent to Equation (2), thus at present the theory is only qualitative.

This study regards the use of a very simple test to contrast the predictions of the standard dielectric theory with the SDM theory. In this study parallel plate capacitors were constructed such that in most cases only ambient laboratory air was between the electrodes. The capacitors were then “immersed” in different media (1) ambient laboratory air, (2) DI water, (3) DI water containing 0.5 wt % NaCl, and (4) DI water containing 5.0 wt % NaCl. According to standard theory the impact of the dielectric properties of material outside the region between the plates is irrelevant, hence all four capacitor “immersed” configurations should operate identically. In contrast, according to SDM theory, the measured capacitance of the test capacitors immersed in water or salt water should be substantially higher than those embedded in ambient laboratory atmosphere. The results, in brief, were that those capacitors immersed in water or salt water had a capacitance at least seven orders of magnitude higher than measured for the same capacitors immersed in air. In fact, for the 5 wt % NaCl case the effective dielectric constant below 1 V was spectacular, more than >10,000,000,000× larger than the same capacitor immersed in laboratory air. Thus, the outcome of the experiments was only consistent with the SDM hypothesis.

This was not the first report of dielectric material outside the volume between the electrodes profoundly impacting performance. All the results reported were consistent with an earlier report from our laboratory, on the behavior of parallel plate capacitors covered with an SDM “gel” outside the volume between the electrodes [2]. As noted earlier, the intent of the present study was to confirm and “generalize” the conclusions reached in the first publication on the topic.

### 4.1. Secondary Findings

Secondary information found in the data included the following: (1) Pure water at short periods, order 1 s (roughly equivalent to a frequency of 1 Hz), had a dielectric constant in excess of 10^7^, as reported elsewhere. (2) Salt does increase the dielectric constant. DI with even low dissolved salt concentrations (ca. 0.5 wt % NaCl) could have remarkably high dielectric values, >10^10^, even for a one second hold time at 10 V. At one second discharge time the difference in the effective dielectric constant between DI, and 5 wt % NaCl in DI, was almost three orders of magnitude. (3) Increasing salt concentration did increase effective dielectric constant. Consistently, a bath of salt with 5 wt % NaCl produced higher capacitance, energy density, etc. values than a bath with 0.5 wt % NaCl. (4) There was evidence of a maximum, or “saturation” value to energy density achievable with salt water dielectric. In this study even as the discharge time was increased, effective dielectric constant remained relatively constant over a range of discharge times from about 1 s to 250 s. (5) Effective dielectric constant values were similar in magnitude to the dielectric constants of the same materials “between the plates”. (6) Finally, in this study it was found that the effective dielectric constant of a dielectric material was always measured to be higher if it were outside the region between the electrodes than if it was placed between the electrodes. All of these secondary findings were only semi-quantitative and more detailed investigation is justified.

Most of these findings were consistent with earlier work on SDM, and expectations developed on the basis of those studies. Indeed, the high effective dielectric constant values for salt water were within an order of magnitude of those published previously for SDM gels on the outside of parallel plate capacitors [2] as well as SDM, in various configurations, “between the electrodes” [3,4,5,6,7,8,9,10,11,12,13].

It is notable that other groups studying the dielectric value of water at low frequency (ca. near 1 Hz) report values of dielectric constant very similar to those reported here [14,15]. Moreover; those teams used other methods, not the constant current method employed herein. Thus, the present results further demonstrate the generality and reliability of the results.

### 4.2. Theory

It is illustrative to compare models of the origin of high dielectric value found in the literature for DI water at low frequency, which is the standard model vs. SDM model. The standard model is that the extremely high dielectric values (ca. 10^7^ at 1 Hz) result from charged species in the water (e.g., OH^−^, H_3_O^+^) forming oppositely charged electric double layers at each electrode [15]. According to the model, at the positive electrode OH^−^ forms a double layer, and at the negative electrode H_3_O^+^. For several reasons it is not at all clear how the remarkably high net dielectric values observed are consistent with that model: (1) In standard supercapacitor models it is assumed the dielectric value of the double layer is in the low double digits [1] at low frequency. (2) This standard model cannot explain why the dielectric constant of water is at least five orders of magnitude greater than solid titanates [1]. Generally some double layer like feature is proposed to explain the dielectric value of solid dielectrics [21,22,23]. (3) The model is not consistent with the fact that voltage is a state property. Given all paths yield equivalent voltage, how does the double layer reduce the voltage for a charge that travels between the electrodes via a path outside the volume between the electrodes?

The SDM model, it is argued, is consistent with all observations, and all laws of physics [28,29]. As explained in more detail elsewhere [1,2,3,4], and illustrated in Figure 9, the theory is based on the field strength and direction of the field generated by a dielectric, placed between electrodes or surrounding the electrodes, “partially cancelling” the quasi dipole field produced by charges on the electrodes. The field generated by polarizing the dielectric will also be quasi-dipole, with a vector direction necessarily opposite at every point in space to that of the “dipole” field generated by charges on the electrodes. Thus, the field produced by the dielectric reduces the electric field produced by the charge on the electrode at every point in space, both between the electrodes and outside the volume between electrodes. As the voltage is the line integral of the field, for any given charge density on the electrodes, that line integral, and concomitantly the voltage, is lowered. Hence, given a constant charge, capacitance (charge/voltage) is increased by the presence of the dielectric.

The SDM model predicts, consistent with the data presented herein, and contrary to the standard model, that dipoles outside the volume between the electrodes will increase capacitance. The dipoles formed in the dielectric reduce the field at every point in space whether the dielectric is in the volume between the electrodes, or outside that volume, a concept completely consistent with the standard E/M theory [30]: The electric field at any point is space is the vector sum of the fields of all charges in the universe. In either geometry the field at all points in space produced by the charges on the electrodes is reduced by the oppositely polarized dipoles of the dielectric. Also, unlike the standard model, there is no need to postulate either a double layer, or a high field region near the electrodes. According to the SDM model the electric field distribution in space is nearly the same for a particular voltage, including the region adjacent to the electrodes, with and without a dielectric [1].

The SDM model applies to all dielectrics, solids, and liquids. Two factors [1,2,3,4] should impact the observed dielectric constant at low frequency; (1) the dipole density in the dielectric and, (2) the dipole length in the dielectric. There is no fundamental difference in the “action” of a solid (e.g., barium titanate) or a liquid dielectric. Both reduce the field created by charges on the electrode at all points in space via the formation of dipoles oriented opposite to the dipole orientation of the electrodes. Indeed, according to SDM theory the underlying physical explanation for the enormous difference in a dielectric constant at low frequency (ca. 1 Hz) between salt water and barium titanate is the length of the dipoles. In barium titanate it is well under 10^−10^ m, and in the salt water it can clearly be even centimeters long [1,2,3,4,5,6,7,8,9,10,11,12,13]. Calculations show that NaCl saturated water has about 30% as many dipoles per volume as barium titanate. The longer dipoles of salt water lead to higher dipole fields at all points in space, hence more significant cancellation of electrode charge produced field, and concomitantly higher capacitance. That is, barium titanate, with far shorter dipoles, according to SDM theory should have far lower dielectric constant than salt water, as observed.

Some features of the data collected for the present work can readily be shown consistent with the SDM model. First, water should be an excellent dielectric at low frequencies because, as suggested elsewhere [15], the dipoles of water molecules align in the presence of an electric field. The structure of water in this condition is not known. Once, aligned, the water molecules will effectively “cancel” the field of the charges on electrodes, leading to extraordinarily high dielectric constants. Second, dissolved ions will further reduce the net field at all points in space by forming an effective large dipole with a length greater than the distance between the electrodes (Figure 9). The magnitude of this dipole may even explain why it was observed that S-DOC outperformed S-DIC as S-DIC dipoles, restricted by the internal volume, are necessarily shorter than those found in the S-DOC configuration. Third, the effect of ion separation should increase with hold time. That is, the longer the hold time, the more charges can travel from elsewhere in the liquid bath to arrive at the proper electrode. In contrast, hold time has virtually no impact on the capacitive behavior of DI. Indeed, there is no need to provide time for ions to travel, only enough time for the water molecule alignment, clearly a far faster process.

Further study of a variety of related topics is arguably justified. What is the impact of salt type? For example, is KCl or NH_3_Cl better than NaCl? Is KOH a better ion source than NaCl? How does the pH of salt-free water impact behavior? Is there a trend in the energy density as a function of inter-electrode distance?

### 4.3. Application

Potential significant applications of the SDM theory supported by these experiments, are: (1) Possible novel energy storage devices, and (2) improved understanding of charge/discharge mechanisms in nerve tissue. Regarding the former: As noted elsewhere, the high dielectric constant value of “salt water” at low frequencies suggest capacitors can be created with higher energy densities than the best batteries. An “ideal” example: A parallel plate capacitor with a gap of 1 micron into which a material of dielectric constant of 1 × 10^10^ and specific gravity of 2 is placed, then charged to 1 V will have an energy density of about 6000 Wh/kg of dielectric. This compares rather well with a lithium ion battery with an energy density of order 150 Wh/kg. Even a less “ideal” capacitor, same dimensions, but a dielectric of only 10^9^, and assuming the dielectric is only thirty percent of the weight, still yields an energy density as good as the best lithium ion batteries. The present work suggests an interesting variation: The SDM dielectric need not be in the space between the electrodes, but in fact can merely “surround” the electrodes. Regarding the latter: One third of the fluid in the body is interstitial water with a relatively high Na^+^ ion concentration. The present work suggests the capacitance of any “solvated” circuit, such as a circuit of neurons, will be impacted by the effective dielectric constant of the surrounding fluid. The present results suggest the dielectric constant of the “salt water” in the body is far higher than previously believed. Thus, the capacitance and charge stored in “biological circuits”, even the roll of ions in interstitial media, may need to be reconsidered.

## 5. Conclusions

All data was consistent with the central postulate of SDM theory: Dielectric material on the outside of a parallel plate capacitor is as effective at increasing capacitance, energy density, and power density as the same dielectric material between the electrodes. In contrast, all data was inconsistent with the standard model of dielectrics applied to parallel plate capacitors: As per Equation (2), only the dielectric material between the electrodes plays a role in determining capacitance, energy, and power density. Thus, the data in this paper suggests the theory of dielectrics presented in standard textbooks [20,21,22] should be reconsidered.

## Figures and Tables

**Figure 1 materials-12-02033-f001:**
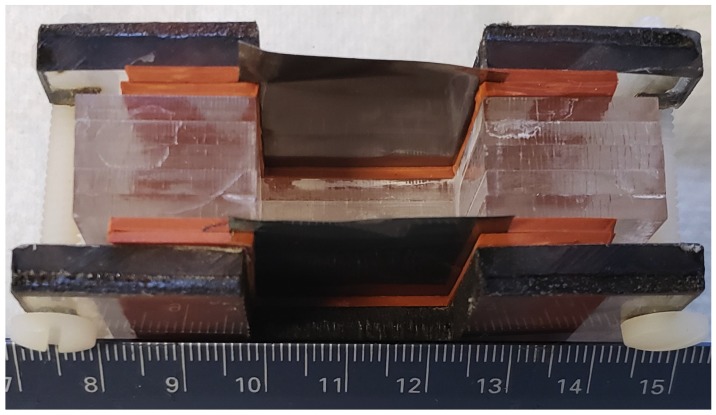
Standard 20 mm capacitor—this capacitor was constructed from two 3 cm × 3 cm × 0.1 mm Ti sheet electrodes, separated by 20 mm. The body of the capacitor was composed of Mylar sheets, and the electrodes were held in place by waterproof rubber (red) gaskets. Plastic screws were used instead of metal to reduce corrosion during immersion experiments. In the configuration shown, after a charge to 10 V, the measured discharge time for the lowest allowed programmable current, 1 × 10^−5^ amps, was 0.0005 s. This value should be compared to the discharges shown in Figure 2. The only difference: The capacitor shown above was partially submerged in distilled water. According to standard theory no difference in discharge time should be observed for submerging a capacitor in any fluid. Note: Unlike most modern studies of the dielectric properties that employ micron-scale devices, herein a multi-centimeter device (see ruler) was used.

**Figure 2 materials-12-02033-f002:**
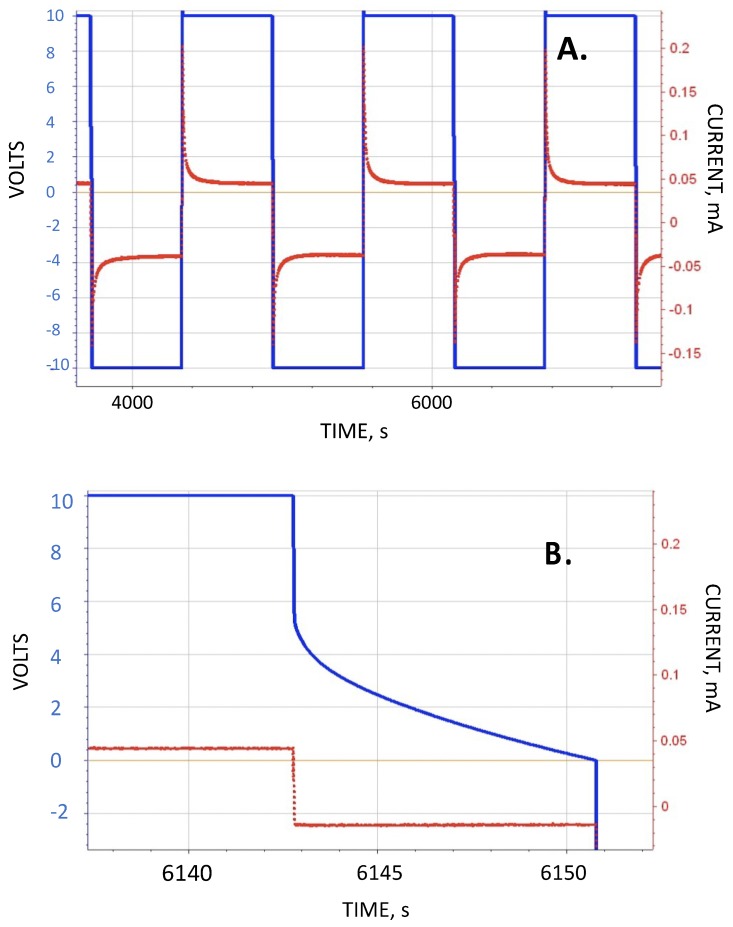
Discharge of DI-DOC for a 20 mm separation capacitor. (**A**) Four positive (10 V to 0 Volts) and three negative (−10 V to 0 V) voltage cycles shown. The hold time at maximum voltage was 600 s, and the constant current discharge current was 0.02 mA. (**B**) An expansion of one of the positive discharges shows that the discharge from 1 V to 0 V took approximately 3 s. The computed capacitance (Equation (1)) below one volt, 6 × 10^−5^ F, was about eight orders of magnitude higher than that computed for the same capacitor sitting on the lab bench. (Lines: Red current, blue voltage).

**Figure 3 materials-12-02033-f003:**
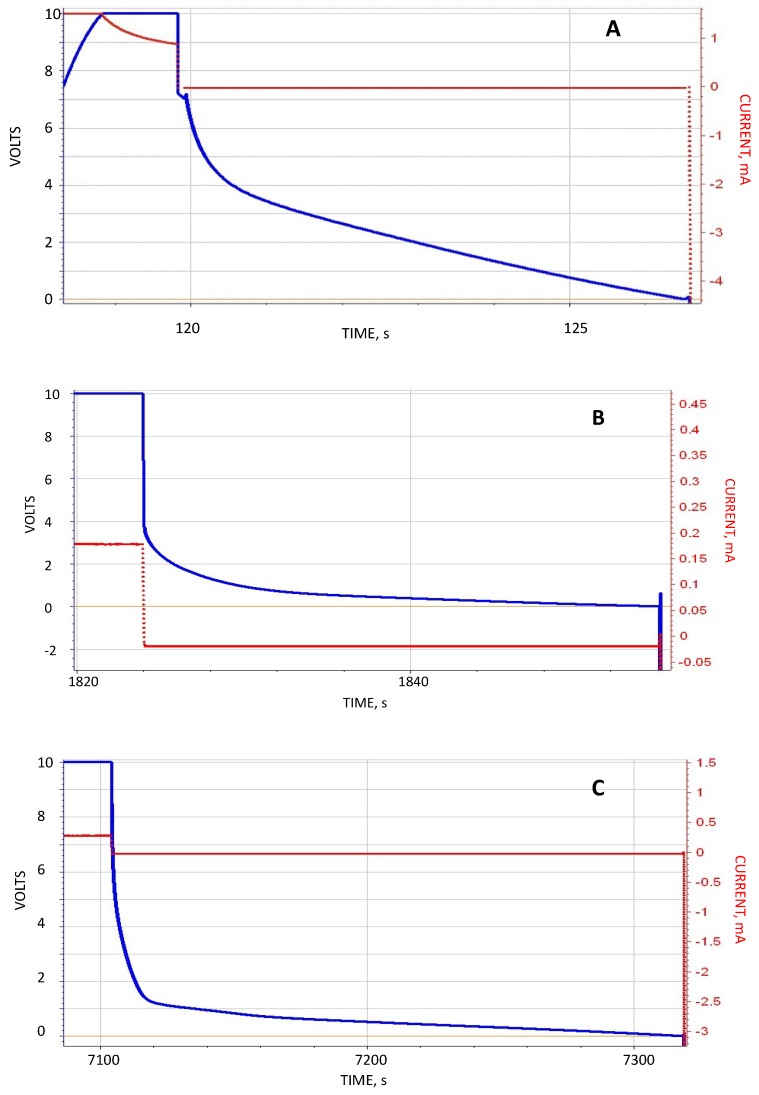
Impact of the hold time for dielectric, 0.5 wt % NaCl in deionized water (DI), only on the outside 20 mm capacitor. (**A**) Shown: Discharge, current 0.02 mA, following a hold time of 1 s at 10 V. The morphology of the discharge behavior for S-DOC (shown) and DI-DOC is very similar. (**B**) Shown: Discharge, current 0.02 mA, following a hold time of 200 s at 10 V. The discharge time (~30 s) was approximately 5× longer than that observed in (A). (**C**) Shown: Discharge, current 0.02 mA following a hold time of 600 s at 10 V. The discharge time (~210 s) was nearly 35× longer than that observed in (A).

**Figure 4 materials-12-02033-f004:**
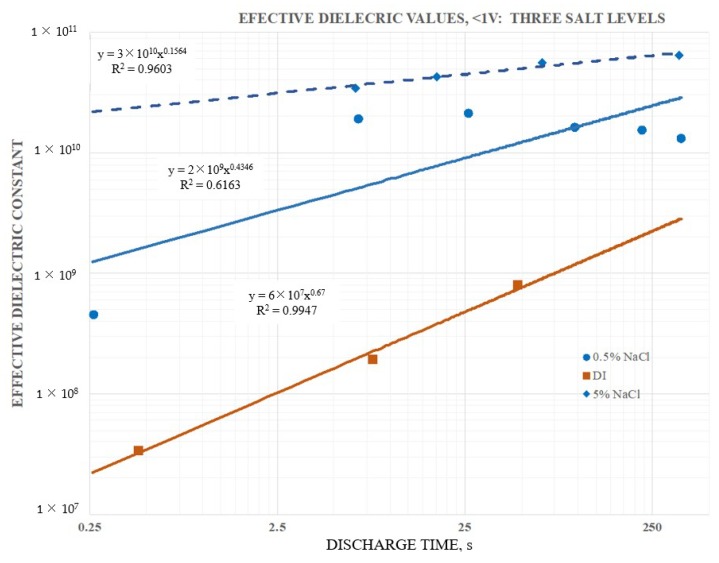
Effective dielectric constant as a function of the salt concentration. The three curves, based on capacitance below 1 V, were obtained with the 20 mm capacitor and were all for a super dielectric outside/ambient air dielectric inside configuration, all based on a program of charging to ±10 V and holding for 200 s at ±10 V. As the salt concentration increases, the effective dielectric constant does. It was also clear that the effective dielectric constant for both the 0.5 and 5 wt % NaCl samples was nearly independent of discharge time for discharge times longer than ~5 s. In particular, the 0.5 wt % NaCl data could not be fitted with a power law.

**Figure 5 materials-12-02033-f005:**
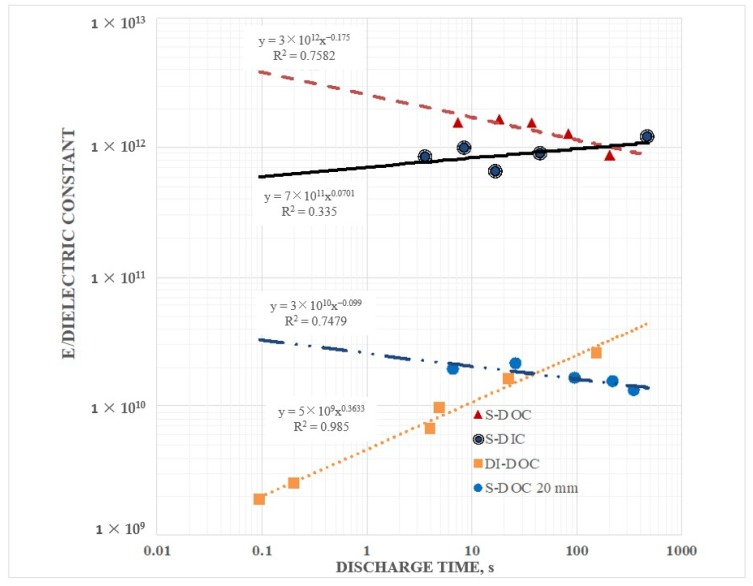
Values for a 6 mm capacitor—Shown are the effective dielectric values below 1 V for the S-DOC and DI-DOC configurations (0.5% NaCl, 10 V charge, 200 s hold) for the 6 mm capacitor. Shown, for comparison, is also the S-DOC for the 20 mm capacitor, same operating parameters. It is notable that the power law fits were imperfect, indicating that extrapolation of the curves was not necessarily valid.

**Figure 6 materials-12-02033-f006:**
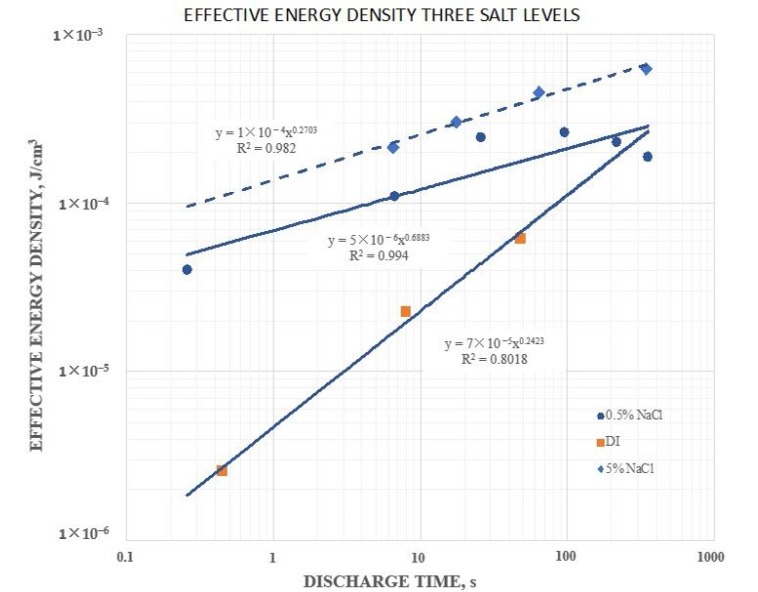
Effective energy density as a function of the salt concentration. The three curves, based on the full voltage discharge, were obtained with the 20 mm capacitor and were all for a super dielectric outside/ambient air dielectric inside configuration, all based on a program of charging to ±10 V and holding for 200 s at ±10 V. As the salt concentration increased, the effective energy density also increased. It was clear that the effective energy density for both the 0.5 and 5 wt % NaCl samples was more sensitive to discharge times than was the effective dielectric constant.

**Figure 7 materials-12-02033-f007:**
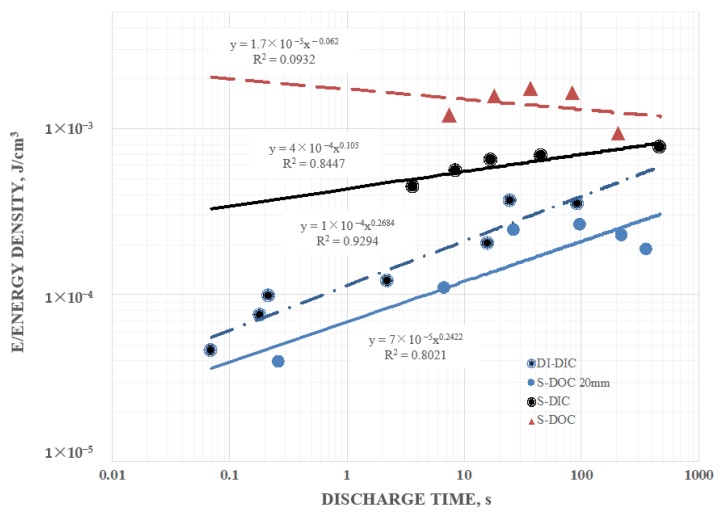
Energy density 6 mm capacitor—the 6 mm capacitor consistently had higher energy density than the 20 mm capacitor in all equivalent configurations. Parameters: ±10 V and holding for 200 s at ±10 V. Note: For the two “DOC” configurations shown the energy density was the effective energy density.

**Figure 8 materials-12-02033-f008:**
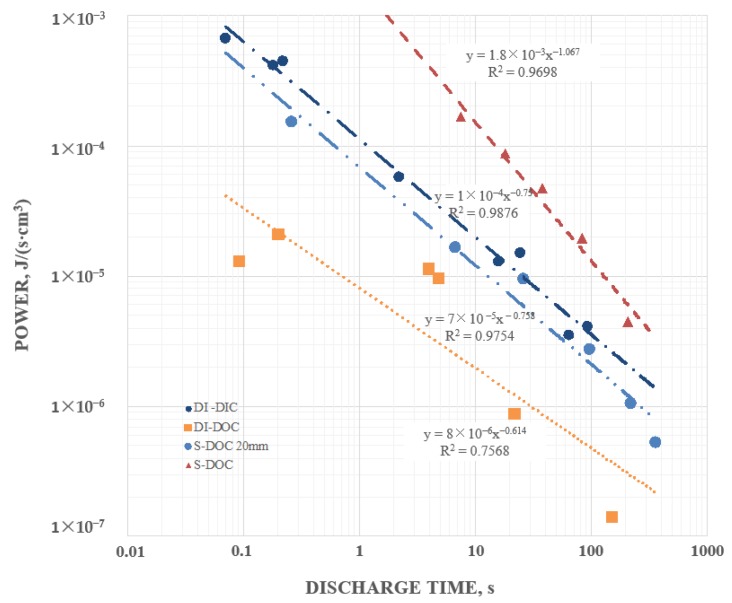
Increases with decreasing discharge time—the highest power, based on energy determined by integration over the full discharge, was found for S-DOC for the 6 mm capacitor and the lowest for the DI-DOC for the 6mm capacitor; however all configurations produced high power and showed the same trend with discharge time.

**Figure 9 materials-12-02033-f009:**
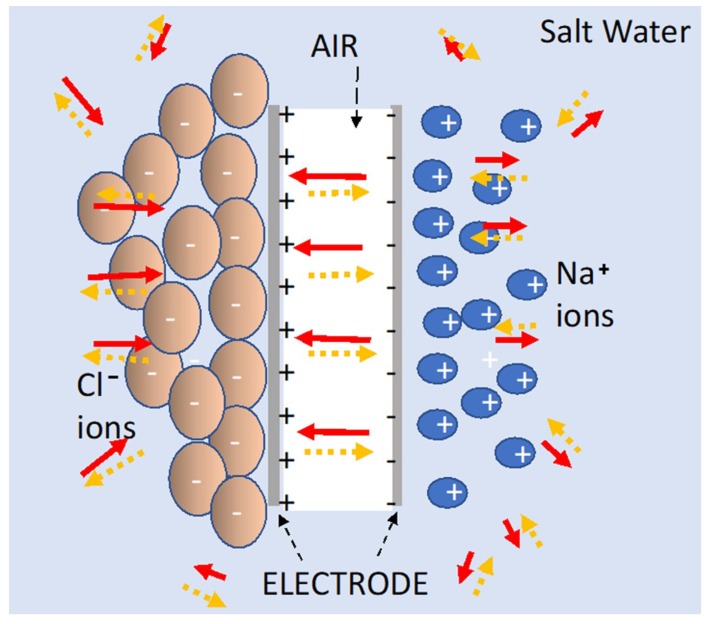
Top view schematic of super dielectric material (SDM) theory for a parallel plate capacitor submerged in a salt solution. In the S-DOC configuration illustrated at all points in space, both “outside” the capacitor and between the electrodes, the field due to charge on the electrodes (solid arrows) is partially cancelled by the field created by the ions (dashed arrows), or by water molecules (not shown) organized into a “liquid crystal” like arrangement.

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
