# Peer review of "Understanding Dielectrics: Impact of External Salt Water Bath"

_materials, 2019, doi:10.3390/ma12122033_

Reviewer 1 Report

The article presents puzzling observations, however, the explanations seem to assume pure dielectric contributions, where perhaps other effects may also be involved. The information provided is not sufficient to rule out alternative explanations. This has limited my ability to rate various aspects of the manuscript.

For example, in Figure 2, the voltage at the beginning of the "charge" is very flat and held at 10V at a current of about 0.04mA. This means the system has a net resistance of about 250,000 ohm. In my opinion this value is too low for air. In Figure 3, where NaCl was added, this charge current is 0.4 mA, one order of magnitude larger than the previous case. Therefore, the possibility of ionic transport from the bath to the electrodes may not be completely ruled out, without additional checks or information. In addition, Are the surfaces of the Titanium electrodes undergoing some electrochemically induced subtle modification at 10V and 0.04mA or 0.4 mA current? Is the high impedance of the measured system interfering with the ability of the Biologic equipment to control both voltage and current? Has the measurement been checked with separate meters? The attribution of measured values to dielectric constants does not seem sufficiently justified. 

Author Response

Response to Reviewer 1:

Comment 1: The article presents puzzling observations, however, the explanations seem to assume pure dielectric contributions, where perhaps other effects may also be involved. The information provided is not sufficient to rule out alternative explanations. This has limited my ability to rate various aspects of the manuscript.

Response:  There can always be another explanation in science, because the ‘real world’, as opposed to a man-invented world (e.g. mathematics), is-what-it-is.   We can proof mathematical theories, but not models of the natural world.  For the natural world we propose models, then devise experiments to test them.  A theory becomes untenable when the predictions of the model are shown to inconsistent with observation.  The more tests a model passes, the more acceptable the model….but it is never proven. In this manuscript, the behavior predicted and the behavior observed matched. It joins a corpus of existing tests, all successfully passed, of the SDM model. So, the model remains tenable, not proven.

    There could be alternatives to Newtonian ‘gravity’ model, to the Big Bang Theory, to String Theory.  It is not the string theorists responsibility to go through all the alternatives, etc.

  The very first paragraph in our paper explains what was done:

In this paper a novel experiment was conducted to test further a new theory of dielectrics, the so-called super dielectric material (SDM) theory. The experimental design of this work was intended to provide a contrast between conventional dielectric theory, as presented in physics texts, and SDM theory. That is, the experiments were designed such that the outcome could only be consistent with one of these theories.

An very familiar  example of The Scientific Process described in the paper is test to demonstrate the viability of General Relativity.  The standard theory  (Newtonian Gravity model), and the only existent theory other than SDM, indicated light would not ‘bend’ as it went around the sun.  GR theory predicted it would.  Measurement was only consistent with the GR model. …And no other models were on the table. But, that historic  singular success does not preclude others postulating new theories!

The point:  We follow standard scientific protocol.  If a model passes a ‘predictive test’ it remains viable.  Others are invited to invent other models, necessarily consistent with all existing data, and test them! 

 The results are NOT puzzling. They are expected.

  And I could go on and on…

Comment 2: For example, in Figure 2, the voltage at the beginning of the "charge" is very flat and held at 10V at a current of about 0.04mA. This means the system has a net resistance of about 250,000 ohm. In my opinion this value is too low for air.

Response-   All measurement devices have some ‘error’. As I now note in the text, one can measure the same tiny current between the two galvanast electrodes with nothing attached to them.  It is some kind of minimal  internal current leak.   Getting bogged down on the control data is a mistake here.  There is no point arguing about the size of the head of the ‘control’ pin.  It is very, very small.  Indeed, using the ‘head of a pin’ analogy:  Assuming the ‘control air dielectric’ pin has a diameter of 1mm, a ‘high dielectric pin head’ 108larger would have a diameter of 10 meters! That is, the control data just shows that the capacitor has virtually no capacitance in air.   

 The discharge time, given the smallest allowed discharge current, for the galvanostat connectors simply placed just above the bench in ambient conditions and that obtained when the electrodes are connected to the capacitor in the AIR configuration are the same. The charging current shows the same pattern as well.  This indicates that the galvanostat is not able to measure discharges that occur more rapidly than 5 10-4 V/s and the current an instrument leakage minimum.

Comment 3 Are the surfaces of the Titanium electrodes undergoing some electrochemically induced subtle modification at 10V and 0.04mA or 0.4 mA current? Is the high impedance of the measured system interfering with the ability of the Biologic equipment to control both voltage and current? Has the measurement been checked with separate meters? The attribution of measured values to dielectric constants does not seem sufficiently justified.

Response-  See Response to the first comment.

 And we do not claim ONE experiment ‘proves’ the model.  The goal, standard scientific protocol,  is to add more and more evidence that the model correctly predicts behavior.  So far there is not a shred of data inconsistent with the model from ANY source.  And now there is data showing the standard model is NOT consistent with observed data

We have re-written the third paragraph of the Intro to make this point.  To wit:

There are some inherent predictions of the SDM model. One example is the prediction that the effectiveness of a dielectric is the product of the length of charge separation within it (dipole length), and the density of charges (dipole density).  This was tested and found accurate in earlier work [3,4].  Other work showed that, as predicted by the model, high dielectric constants would be found for salt water saturated fabric [5], for salt water saturated nano- tubes on the surface of anodized titania [6], for porous solids saturated with various salts [7,8], for fumed silica gels containing salt water [9], for water saturated with salts other than NaCl such as KOH and NH3Cl [10].  Other studies show that both metal and carbon can be used as the electrode material [2], and for non-aqueous polar  fluids such as DMSO containing dissolved salts, etc [1-13]. Each study adds to the corpus of data supporting the SDM hypothesis.

Commment 4- For example, in Figure 2, the voltage at the beginning of the "charge" is very flat and held at 10V at a current of about 0.04mA. 

Response-  See response to Comment 2.

Comment 5-This means the system has a net resistance of about 250,000 ohm. In my opinion this value is too low for air. 

Response-  This is a misunderstanding of the constant current method, and leakage currents.  See response to Comment 2.

If the reviewer can find, in the literature, a single example of the use of the data in the manner suggested to obtain ‘resistance’ of dielectric materials, we will reconsider this response.

Comment 6- In Figure 3, where NaCl was added, this charge current is 0.4 mA, one order of magnitude larger than the previous case. Therefore, the possibility of ionic transport from the bath to the electrodes may not be completely ruled out, without additional checks or information. In addition, Are the surfaces of the Titanium electrodes undergoing some electrochemically induced subtle modification at 10V and 0.04mA or 0.4 mA current? Is the high impedance of the measured system interfering with the ability of the Biologic equipment to control both voltage and current? Has the measurement been checked with separate meters? The attribution of measured values to dielectric constants does not seem sufficiently justified. 

Response-  The reviewer implies that increased ion transport would increase the discharge time?  Indeed, in air we can assume virtually no ion transport.   So why would increasing ion transport in liquids, as suggested by the reviewer,  increase discharge time?  Would it not simply create a short circuit, decreasing discharge time yet further? And note, as described in this article and others, with the end of each discharge  we switch the identify of the electrodes(e.g. anode becomes cathode), and find it makes no difference.  Want to try that with a battery? 

    The reviewer is entitled to repeat the measurements, using any method.   He is not entitled to demand rigor way beyond that used by others.  Separate meters?  Is the reviewer able to cite a single example in the literature of capacitance measurement where ‘separate meters’ were employed?  Will the reviewer send $20 K for a second meter, please? And the measurements made herein are  completely consistent with a host of other measurements on SDM based capacitors made with both RC time constant (3-9) and constant current protocols (2, 10-13). In any event, we have now added this absolutely true statement:

(Notably, the device is regularly tested by using it to measure the marked capacitance of both commercial supercapacitors and electrostatic capacitors. The agreement with nominal capacitance is always excellent.)

For example, I regularly teach a class on capacitors and batteries to US Naval officer graduate students. These students, in several groups, re-test the galvanostat as described above and report in detail.  Yes, the galvanastat readings are accurate.  Also, just for the record, we have now submerged in baby oil one of the same well used capacitors employed in the paper. And, lo and behold, it shows no change in capacitance relative to submergence in air! As baby oil is non-polar, this is expected according to SDM theory.

      The reviewer does not suggest a reasonable alternative mechanism for why submerging a capacitor in water, or salt water, would dramatically change the measured capacitance.  Making reference to unidentified, undescribed, ‘Maxwell Demons’ is not accepted scientific practice.   And, he cannot cite an alternative in the literature, because none exists.  Moreover; the observation is completely predicted by SDM.        

    We stand behind our expertise.  This author has reviewed all methods of determining capacitance (ref. 1) and demonstrated that at low frequency the constant current method is clearly the best, and as noted, it works!   We have also used the RC time constant method to test (e.g. Refs 6-9)  SDM and get the same magnitude dielectric values obtained using the constant current method.  And, we had a colleague measure the dielectric properties of some of our SDM using an Impedance Spectrometer.  The result:  Even higher dielectric values than we measured…..

Reviewer 2 Report

In this work, the author demonstrated that the dielectric material on the outside of a parallel plate increases capacitance, energy density, and power density. The author needs to address the following issues before I consider for publication.

1) In figure 1 what does the ruler indicates? They should write in the caption.

2) On page 5, from line 180 to 187, multiple signs are missing in reporting some values. Such as. 5 10-4 V/s.

3) In figure 2, the author should clearly indicate what does blue and red color lines represents. 

4) In the results and discussion section, the referencing is not properly added. In line, 296 the author mentioned the references from 1 to 13. Similarly, in line 359, the author cited references from 3 to 13. They should be specific in citing articles. This is not a proper way of citing articles.

5) in the last line of the conclusion from line 458 to 460 the author wrote that "

Thus, the data in this paper suggests the theory of dielectrics presented in standard textbooks [20-22] should be reconsidered." This is a strong statement. They cannot claim such a strong statement by performing only a few experiments. I am suggesting to remove the line.

Author Response

Response to Reviewer 2:

In this work, the author demonstrated that the dielectric material on the outside of a parallel plate increases capacitance, energy density, and power density. The author needs to address the following issues before I consider for publication.

Comment 1: 1) In figure 1 what does the ruler indicates? They should write in the caption.

Response The was added to the end of the caption: Note: unlike most modern studies of the dielectric properties that employ micron-scale devices, herein a multi-centimeter device (see ruler) was used.

Comment 2) On page 5, from line 180 to 187, multiple signs are missing in reporting some values. Such as. 5 10-4 V/s.

Response: Corrected.

3) In figure 2, the author should clearly indicate what does blue and red color lines represents. 

Response: Added at end of caption:  Lines: Red current, blue voltage

 Comment 4) In the results and discussion section, the referencing is not properly added. In line, 296 the author mentioned the references from 1 to 13. Similarly, in line 359, the author cited references from 3 to 13. They should be specific in citing articles. This is not a proper way of citing articles.

 Response:  The third paragraph in the Introduction ‘breaks down’ these references.  To wit:

There are some inherent predictions of the SDM model. One example is the prediction that the effectiveness of a dielectric is the product of the length of charge separation within it (dipole length), and the density of charges (dipole density).  This was tested and found accurate in earlier work [3,4].  Other work showed that, as predicted by the model, high dielectric constants would be found for salt water saturated fabric [5], for salt water saturated nano- tubes on the surface of anodized titania [6], for porous solids saturated with various salts [7,8], for fumed silica gels containing salt water [9], for water saturated with salts other than NaCl such as KOH and NH3Cl [10]. Other studies show that both metal and carbon can be used as the electrode material [2], and for non-aqueous polar  fluids such as DMSO containing dissolved salts, etc.  Each of the existing SDM  studies [1-13] adds to the corpus of data supporting the SDM hypothesis.

Other ‘groups’ of references are done to make a point.  For example, this one gives the reader the ENTIRE historical list of SDM references, making the point there are not that many, and you can look them all up: 

In contrast to energy density, for all reported SDM based capacitors [1-13], power increases as the discharge time decreases.

Note: This author has ~170 reviewed publications, plus 35 issued patents, been cited more than 5000 times. …. I’ve got this. 

 Comment 5) in the last line of the conclusion from line 458 to 460 the author wrote that "

Thus, the data in this paper suggests the theory of dielectrics presented in standard textbooks [20-22] should be reconsidered." This is a strong statement. They cannot claim such a strong statement by performing only a few experiments. I am suggesting to remove the line.

Response:  Not a few experiments: There are now 13 reviewed publications, and about ten MS theses, full of data consistent with  that statement.   

We are not snowflakes. We can and will take the heat, because what is stated is true.

Reviewer 3 Report

To the Authors,

The article, which is mainly passing information on the conducted works focusing on the behaviour of capacitors in the field of low voltages, indicates the occurrence of interesting effects. However, there is a concern regarding the methodology of research which may significantly influence the outcome. The use of air as a dielectric may pose a danger that it  can possibly be contaminated, or that there may be vapours of conductive gases that could noticeably affect the results. Therefore, I would suggest using synthetic air in tests. The effects observed in the low voltage range can be crucial. For that reason, I would be very cautious with the statement in line 332 – in addition, Figure 4 does not show such a high increase as presented in this line but only about 3 decades. Furthermore, the article requires considerable editorial changes.

Below please note my remarks:

a)      No abbreviations shall be used in the abstract

b)      The article is not written in accordance with the MDPI text standard

c)      The use of tags such as i] in the text is misleading,

d)     Line 58 not ~ 1 Volt but ~ 1 V,

e)      Formula 1 and its description should be corrected (current is missing)

f)       Unfortunately, the figures require substantial changes:

- are not in accordance with the recommendations, e.g. A and not (a) Fig 2, Fig3, etc.,

- captions should be directed in such a way that their reading does not require rotation of the figure,

- it is necessary to standardize the units used in the axis as they are different, e.g. Fig. 2 Fig. 4, Fig. 6

- errors in captions, e.g. Fig. 2,

- there is an unidentified error while printing page 13; it is related to Fig 9

- the size of the fonts used in the figures is very diverse, they shall be unified.

 At the same time, according to the reviewer, it would be worth including in the paper a note that the article was presented in the preprint version on:

 https://www.preprints.org/manuscript/201903.0289/v1/download

 Author Response

Response to Reviewer 3:

The article, which is mainly passing information on the conducted works focusing on the behaviour of capacitors in the field of low voltages, indicates the occurrence of interesting effects. However, there is a concern regarding the methodology of research which may significantly influence the outcome. 

Comment 1: The use of air as a dielectric may pose a danger that it  can possibly be contaminated, or that there may be vapours of conductive gases that could noticeably affect the results. Therefore, I would suggest using synthetic air in tests. 

Response 1:  Consistent with the SDM theory, all gases, every single one, have the same dielectric constant within 5%.  And that value is within 5% of the dielectric constant of vacuum. 

Conductive gases? 

All sources same the same thing: “At atmospheric pressure, air and other gases are poor conductors (Insulators) of electricity…. they don't have any free electrons to carry current”

Only plasmas are good conductors.  As the author of dozens of plasma papers, I can say with absolute certainty, there is no plasma in our system.

 Comment 2: The effects observed in the low voltage range can be crucial. For that reason, I would be very cautious with the statement in line 332 

Response:  This is the statement in question: 

In fact, for the 5 wt% NaCl case the effective dielectric constant below 1 V was spectacular, more than > 10,000,000,000 X larger than the same capacitor immersed in laboratory air. Thus, the outcome of the experiments is only consistent with the SDM hypothesis.

It is completely true.  

Please note the following relevant facts:

1.    The only true competition to SDM based capacitors (Novel Paradigm Supercapacitors, NPS) for low frequency energy storage are EDLC (aka supercapacitors).  NPS and EDLC operate in the same voltage region (ca.<1.5V).  Thus, the method used, and the parameter range for which conclusions were drawn, are appropriate for the intended applications.

2.    Most modern capacitor studies are performed using Impedance Spectroscopy (IS).  This method only measures behavior over the range +/- 15 mV. For this reason, it was judged an inappropriate method for study of SDM. Should all studies of capacitance based on the IS method be discounted?

All of this information/reasoning can be found in Ref. 1 of the paper. 

Comment 3:– in addition, Figure 4 does not show such a high increase as presented in this line but only about 3 decades. Furthermore, the article requires considerable editorial changes. 

 Response: Figure 4 only shows the capacitor immersed in salt water or DI water.  It does not show the capacitor immersed in laboratory air.  True, the difference between DI water and water with 5wt% NaCl is ‘only’ ~4 orders of magnitude. However;  the basis of the comment cited above is the difference between the 5wt% NaCl case and laboratory air (Clearly stated!), and that difference is 10,000,000,000 X. No error. 

Below please note my remarks:

Comment 4: No abbreviations shall be used in the abstract

Response: Employed the standard style guide:

 Avoid acronyms in the abstract unless the acronym is commonly understood and used multiple times in the abstract. If an acronym is used in the abstract, it must be spelled out (defined) in the abstract, and then spelled out again the first time it is used in the body of the paper.

 DI is commonly understood, spelled out, and used multiple times in the Abstract.

Comment 5:   The article is not written in accordance with the MDPI text standard 

Response:  Corrected. 

Comment 6:    The use of tags such as i] in the text is misleading,

Response:  Really?

d)     Line 58 not ~ 1 Volt but ~ 1 V,

Response: Corrected.

Comment 7:      Formula 1 and its description should be corrected (current is missing)

Response: Corrected.

 Comment 8: f)       Unfortunately, the figures require substantial changes:

- are not in accordance with the recommendations, e.g. A and not (a) Fig 2, Fig3, etc.,

Response:  Corrected.

Comment 9:- captions should be directed in such a way that their reading does not require rotation of the figure,

Response:  Corrected.

Comment 10: - it is necessary to standardize the units used in the axis as they are different, e.g. Fig. 2 Fig. 4, Fig. 6

Response:  Same units in all figures that are equivalent. For example,  Figs 2 and 3 show ‘equivalent data’ and same axes. Figures 4 and 5 show ‘equivalent data’ and have same axes.

Comment 11:- errors in captions, e.g. Fig. 2,

Response:  Could find no errors in the captions. 

Comment 12:- there is an unidentified error while printing page 13; it is related to Fig 9

Response:  New printer needed?

Comment 13: - the size of the fonts used in the figures is very diverse, they shall be unified.

Response:  The same size fonts were used repeatedly in the originals.  The figures cover different parameter ranges, making the absolute original sizes different.  Once inserted in the text they must be ‘expanded’ or ‘condensed’ to fit the document.  This gives the appearance of different size fonts.  This is very, very difficult to overcome. 

 Comment 14: At the same time, according to the reviewer, it would be worth including in the paper a note that the article was presented in the preprint version on:

 https://www.preprints.org/manuscript/201903.0289/v1/download

Response:  Given that MDPI invited/instigated the ‘pre-print’, I suppose they must have a policy on citation. What is that policy?

Round  2

Reviewer 1 Report

The authors have disregarded all my previous comments. There are indications of ionic currents interacting with the electrodes. Deposition of species on the electrodes can not be ruled out. All phenomena is discussed in terms of dielectric effects. In my opinion ionic current effects have not been adequately ruled out.

Author Response

General comment:  Many scientists are unable to accept the verdict of the scientific process if the outcome challenges the standard paradigm.  These individuals are not suitable scientific reviewers, because they act as 'enforcers of dogma'.  This is a religious, not a scientific, avocation.  

Comment 1: The authors have disregarded all my previous comments.

Response:  I responded to ALL of his comments in detail. If the reviewer has concerns with my detailed responses, he must present them.  The authors cannot respond  to the above comment, because there is nothing to respond to.

Comment 2: There are indications of ionic currents interacting with the electrodes. Deposition of species on the electrodes can not be ruled out.

Response:  There are no indications of 'ionic currents interacting with electrodes.'

Comment 3: All phenomena is discussed in terms of dielectric effects.

Response:  Correct, except for the grammar.

Comment 4: In my opinion ionic current effects have not been adequately ruled out.

Response:  The reviewer is opining there is a 'battery' process responsible for all observations.

Reasons this cannot be:

A. Batteries are essentially devices for converting chemical energy into electrical energy.  They have distinct anodes (electrons and ions leave) and distinct cathodes.  In our study we  reversed, over and over again, the identity of the  electrodes, equivalent to reversing identities of anode and cathode.  Discharge behavior was unchanged!  This cannot happen with a chemical system with a distinct anode and cathode (battery).   

B.  In constant current discharge mode for a battery, the discharge voltage is virtually constant until the chemical process is nearly exhausted.  Not observed here.

C.  All legitimate models of batteries provide detailed chemical reaction schemes that must match observed voltages with Delta G of the reaction.  For example, the double sulfate model for lead acid batteries. For Li ion batteries the model is that graphite intercalated Li ions move through an electrolyte to CoMo cathode structures where they react. In the manuscript we report energy release in a system consisting of titanium and DI water.  What is the proposed chemistry?  Without a specific proposed chemistry, the 'battery hypothesis' is merely a call to Maxwell's Demons.  It has no form and no merit.  

Reviewer 2 Report

I am accepting the paper for publication in the current form.

Note to the corresponding author:

"We can and will take the heat, because what is stated is true. "

If you want to take the heat then next time go to some high impact factor journals. You will feel the real heat. Good Luck.

Author Response

I am accepting the paper for publication in the current form.

Note to the corresponding author:

"We can and will take the heat, because what is stated is true. "

If you want to take the heat then next time go to some high impact factor journals. You will feel the real heat. Good Luck.

Response- Thank you.  We checked the manuscript for spelling one final time....